# Contribution of Genetic Test to Early Diagnosis of Methylenetetrahydrofolate Reductase (MTHFR) Deficiency: The Experience of a Reference Center in Southern Italy

**DOI:** 10.3390/genes14050980

**Published:** 2023-04-26

**Authors:** Ferdinando Barretta, Fabiana Uomo, Simona Fecarotta, Lucia Albano, Daniela Crisci, Alessandra Verde, Maria Grazia Fisco, Giovanna Gallo, Daniela Dottore Stagna, Maria Rosaria Pricolo, Marianna Alagia, Gaetano Terrone, Alessandro Rossi, Giancarlo Parenti, Margherita Ruoppolo, Cristina Mazzaccara, Giulia Frisso

**Affiliations:** 1Department of Molecular Medicine and Medical Biotechnology, University of Naples Federico II, 80131 Naples, Italy; barretta@ceinge.unina.it (F.B.); uomo@ceinge.unina.it (F.U.); d.dottorstagna@studenti.unina.it (D.D.S.); margherita.ruoppolo@unina.it (M.R.); 2CEINGE Advanced Biotechnologies Franco Salvatore, 80131 Naples, Italy; albano@ceinge.unina.it (L.A.); crisci@ceinge.unina.it (D.C.); fisco@ceinge.unina.it (M.G.F.); gallo@ceinge.unina.it (G.G.); 3Metabolic Diseases Unit, Department of Translational Medical Science, Section of Pediatrics, University of Naples Federico II, 80131 Naples, Italy; simona.fecarotta@unina.it (S.F.); verde-alessandra@libero.it (A.V.); manny.alagia@gmail.com (M.A.); gaetano.terrone@unina.it (G.T.); alessandro.rossi@unina.it (A.R.); parenti@unina.it (G.P.); 4Centro Nacional de Investigaciones Cardiovasculares (CNIC), 28029 Madrid, Spain; mariarosaria.pricolo@cnic.es

**Keywords:** methylenetetrahydrofolate reductase (MTHFR) deficiency, hereditary metabolic diseases, newborn screening (NBS) for inborn error of metabolism, genetic test, homocysteine, methionine

## Abstract

Background: the deficiency of 5,10-Methylenetetrahydrofolate reductase (MTHFR) constitutes a rare and severe metabolic disease and is included in most expanded newborn screening (NBS) programs worldwide. Patients with severe MTHFR deficiency develop neurological disorders and premature vascular disease. Timely diagnosis through NBS allows early treatment, resulting in improved outcomes. Methods: we report the diagnostic yield of genetic testing for MTHFR deficiency diagnosis, in a reference Centre of Southern Italy between 2017 and 2022. MTHFR deficiency was suspected in four newborns showing hypomethioninemia and hyperhomocysteinemia; otherwise, one patient born in pre-screening era showed clinical symptoms and laboratory signs that prompted to perform genetic testing for MTHFR deficiency. Results: molecular analysis of the *MTHFR* gene revealed a genotype compatible with MTHFR deficiency in two NBS-positive newborns and in the symptomatic patient. This allowed for promptly beginning the adequate metabolic therapy. Conclusions: our results strongly support the need for genetic testing to quickly support the definitive diagnosis of MTHFR deficiency and start therapy. Furthermore, our study extends knowledge of the molecular epidemiology of MTHFR deficiency by identifying a novel mutation in the *MTHFR* gene.

## 1. Introduction

Hereditary metabolic diseases are heterogeneous congenital disorders, caused by deficiency of a specific metabolic pathway, often leading to early neonatal mortality due to irreversible damage in several organs and apparatus, with neuro-motor and cerebral disabilities. Early diagnoses may have a crucial role in the efficacy of the treatment in order to avoid clinical injury [1,2,3].

Deficiency of 5,10-Methylenetetrahydrofolate reductase (MTHFR) is a very rare and severe hereditary defect of folate metabolism with autosomal recessive inheritance, and is included in newborn screening (NBS) programs for the identification of inborn error of metabolism in many countries [4,5,6,7]. MTHFR deficiency (OMIM#607093) appears typically in the neonatal period, with severe neurological signs, recurrent apnea, microcephaly, encephalopathy, hypotonia, and seizures, although late onset forms of MTHFR deficiency showing seizures, cognitive impairment, neuropathy and psychiatric disorder are also known [8,9,10,11]. The enzyme MTHFR (EC 1.5.1.20) catalyzes the NADPH-dependent conversion of 5,10-methylenetetrahydrofolate (5,10-MTHF) to 5-methyltetrahydrofolate (5-MTHF), using Flavin Adenine Dinucleotide (FAD) as a cofactor. The 5-MTHF serves as donor of the methyl group in the remethylation of homocysteine to methionine, catalyzed by methionine synthase (MS) which uses vitamin B12 as a cofactor. Methionine is then converted into S-adenosyl methionine (SAM), a universal methyl donor, essential for DNA and RNA methylation and for the synthesis of creatine, phospholipids and many neurotransmitters [12] (Figure 1). Accordingly, MTHFR deficiency is associated with a defect in folate processing, resulting in hypomethioninemia and hyperhomocysteinemia in affected patients [13]. Methionine is the NBS primary marker which highlights the MTHFR deficiency.

Mutations in the *MTHFR* gene (NM_005957.5; ID: 4524; 1p36.22) involving both the N-terminal catalytic and C-terminal regulatory enzyme domains have been associated with MTHFR deficiency [14,15,16]. About 200 different pathogenic variants have currently been described in the *MTHFR* gene (Figure 2), and only a few hundred patients have been reported worldwide [8,17,18]. In addition, two common MTHFR variants, namely c.665C>T p.(Ala222Val, rs1801133) and c.1286A>C p.(Glu429Ala, rs1801131), affecting amino acid residues localized within the catalytic and regulatory domains of the MTHFR enzyme, respectively, have been associated with a mild form of MTHFR deficiency and are known genetic risk factors for thrombophilia [19]. The c.665C>T polymorphism encodes a thermolabile enzyme (65% of residual enzymatic activity for heterozygotes and 30% for homozygotes), which is correlated to a slightly higher level of homocysteine. The c.1286A>C polymorphism in compound heterozygosity with c.665C>T results in a functional impact similar to that caused by the c.665C>T homozygous genotype [16,20].

To date, few studies report the prevalence of 5,10-methylenetetrahydrofolate reductase deficiency on expanded newborn screening in Europe. In 2018, Gramer et al. published the results of the study “Newborn Screening 2020”, started at the Newborn Screening Center, Heidelberg, in August 2016 [21]. Among the 68,418 participants in the study, one newborn was affected by MTHFR deficit. Later, David et al. published the results of the NBS program in the Czech Republic from 1 January 2010 to 31 December 2017, which covered 888,891 neonates. In this large cohort of screened newborns, no infants were diagnosed as affected by MTHFR deficiency [22]. Finally, NBS conducted in Italy between January 2019 and December 2020 diagnosed three affected neonates among the 806,769 screened globally [7]. However, none of the aforementioned studies discussed the molecular characterization of the patients.

The aim of this study was to analyze the epidemiology of MTHFR deficiency screened by NBS in the Campania region, following the introduction of extended newborn screening (study period: 2017–2022). Particularly, we discuss the key role of the genetic test to achieve the definitive diagnosis of this rare but very severe disease.

## 2. Materials and Methods

Written informed consent to perform genetic analysis and for the publication of data was obtained from the parents, for themselves and for the newborns. All procedures were in accordance with the standards of the Ethics Committees on human experimentation (Institutional and national) and with the Helsinki Declaration [23] and were approved by the local Ethics Committee.

### 2.1. Newborn Screening

Newborn metabolic screening and genetic analysis (see Section 2.2) were performed at CEINGE_Biotecnologie Avanzate Franco Salvatore (Naples, Italy), which is Unique Regional Center for Newborn Screening in Campania and Regional Reference Center of the National Health Service for Clinical Molecular Biology, Laboratory Genetics.

The newborn blood sample was obtained by heel prick and spotted on a Schleicher & Schuell 903 grade filter paper sampling card (Whatman, Dassel, Germany). Blood collection for newborn screening is recommended between 48 and 72 h of life. Dried blood spot samples were delivered daily by courier to the laboratory [24]. The analytic protocol provided for the extraction of amino acids and acylcarnitines identified as biomarkers of the inborn errors of metabolism subject to screening, as institutionalized by Italian law (2016–2017). The analysis was carried out by LC-MS/MS as described before [2,25]. The results of the screening test are considered “positive” if the value of one or more biomarkers is above or below the reference upper or lower cutoff. The laboratory periodically reviews the cutoffs, taking into account the percentiles distribution of each analyte and the values observed in the false positive subjects. The latest analysis of false positives relates to the three-year period 2019–2021. Whole blood and serum were collected and stored between 2 and 8 °C. Fresh urine samples were collected in a test tube and stored frozen. Serum amino acid determination by HPLC was carried out as described before [26].

### 2.2. Genetic Analysis

Genomic DNA was extracted from peripheral blood of the probands and their parents, as previously described [27].

We analyzed probands collected before June 2020 by Sanger sequencing, using polymerase chain reaction amplification of all 12 exons, the exon-intron boundaries and the 5′- and 3′-UTR regions of MTHFR gene. Since June 2020, we performed next-generation sequencing (NGS), using a home-made target panel of 147 genes, which allows genetic diagnosis of the 47 hereditary metabolic disorders subjected to newborn screening in Italy, including MTHFR deficiency. DNA sequencing, variants filtering, and prediction of disease-causing variants were performed as previously described [28]. In the last decade, NGS has proved to be an effective molecular strategy showing good diagnostic sensitivity [29].

The patients’ sequences were compared with the GenBank^®^ reference sequence of MTHFR (NG_013351.1; NM_005957.5). Furthermore, genetic variants detected in the patients were screened, by Sanger sequencing, in their parents, to verify the family segregation.

To search for gene macrodeletions in apparently negative patients, we calculated the diagnostic index (ID) by normalizing the reads number of the target regions (exons of MTHFR gene) to a double dose internal control and then to the media of three controls analyzed in the same batch. The DI for deleted exons is between 0.4 and 0.6 [30].

### 2.3. Bioinformatics Analysis

To predict the possible pathogenicity of the novel detected variant in silico, bioinformatic analysis was performed by using the Alamut Visual Software (Interactive Biosoftware; http://www.interactive-biosoftware.com/, accessed on 14 December 2022) and VarSome: The Human Genomic Variant Search Engine [31] (https://varsome.com, accessed on 14 December 2022). Several studies have shown the efficacy of bioinformatics analysis to verify the pathogenicity of new variants [32,33]. Novel variants were classified according to the American College of Medical Genetics and Genomics (ACMG) guidelines [34].

### 2.4. Homology Modelling

We produced an automated full-length protein structural prediction homology modelling of MTHFR enzyme, using the protein sequence P42898 from Uniprot database, corresponding to the isoform 1 of protein. Structural predictions of MTHFR wild-type (WT) and MTHFR Leu590Arg mutant protein were carried out using I-TASSER (Iterative Threading ASSEmbly Refinement) server, including X-RAY structure of human MTHFR (6FCX PDB) as template (http://zhanglab.ccmb.med.umich.edu/I-TASSER, accessed on 19 December 2022) [35]. To overlap WT and mutant protein structures, the PDB files, obtained from I-TASSER server, were visualized with PyMOL Molecular Graphics System (The PyMOL Molecular Graphics System, Version 2.0 Schrödinger, LLC, New York, NY, USA) [36].

## 3. Results

### 3.1. Clinical and Biochemical Findings

Screening for inborn errors of metabolism was institutionalized in Italy by law between 2016 and 2017 (Law 167/2016, DM 13 October 2016, DPCM 12-1-2017). The DM 13 October 2016 states that the screening program is a system articulated into four main functions (the screening laboratory, the laboratory for confirmatory diagnosis, the clinical centers, the regional coordination/supervision), defines the panel of screening conditions, the timing for specimen collection, the screening methodology, the confirmatory tests and the clinical follow up.

Here we report results of the screening for MTHFR deficiency in Campania Region in the period from 1 January 2017 to 31 December 2022, during which 249,642 newborns were screened at our center.

Neonatal metabolic screening identified four newborns, all in term-born (between 37 and 41 weeks of gestation) and with normal weight (2500–4200 g), showing low methionine and high homocysteine values (as second-tier testing, Table 1). Following NBS results, the four newborns were admitted to the Day Hospital of the Department of Translational Medical Sciences, Section of Pediatrics, University Federico II, Naples (Italy). At the time of the first clinical evaluation, all the newborns showed good general condition and unremarkable neurological examination. The second-level serum tests, performed within one month after birth, showed low methionine and hyperhomocysteinemia in patient 1 and borderline/normal methionine in combination with confirmed hyperhomocystinemia in patients 2, 3 and 4 (Table 1). The acylcarnitine profile, urinary organic acids measurements, base-acid balance, biochemical profile, blood count, and folate and vitamin B12 resulted in the reference range in three out of four patients. In patient 4, vitamin B12 levels were very low, due to maternal B12 deficiency, urinary and plasma methylmalonic acid were consistently increased (Appendix A). In the light of biochemical findings, a severe deficiency of MTHFR was suspected in patients 1 and 3, prompting to require molecular analysis looking for MTHFR pathogenic variants. Molecular analysis of a panel of genes associated with hyperhomocysteinemia, including MTHFR gene, was also required for patients 2 and 4, in order to achieve the definitive diagnosis.

In agreement with the hypothesis of a severe MTHFR deficiency, a therapeutic regimen including high-dose oral 5-methyltetrahydrofolate and betaine was started in two out of four patients identified at the neonatal metabolic screening (Appendix A). For patient 1, an additional hydroxycobalamin supplementation was temporarily added, as serum vitamin B12 levels were unknown. For one patient (patient 4) therapy with parenteral hydroxocobalamin and carnitine was started, based on the evidence of increased methylmalonic acid; a suspicion of maternal B12 deficiency was based on patient’s and mother’s extremely low B12 blood levels, that could be related to the biochemical anomalies, in combination with normal levels of methionine on HPLC amino acids analysis. Detail of therapeutic regiment is shown in the Appendix A. During the same period, a 12-year-old girl came into the observation of pediatricians due to neurological regression with progressive loss of cognitive and motor skills, appearance of numbness and hypertonicity of the lower limbs and mild intellectual disability. She had not been screened by NBS (born in 2009); metabolic investigations showed marked hyperhomocysteine associated with low levels of plasmatic methionine (Pt 5 in Table 1), which prompted pediatricians to request genetic testing for MTHFR deficiency and to start therapy with 5-methyltetrahydrofolate (45 mg/die), betaine (250 mg/kg/die) and vitamin B complex.

### 3.2. Molecular Analysis of the MTHFR Gene and In Silico Evaluation of the Novel Variant

Sequencing of the coding region of MTHFR gene revealed a genotype compatible with the definitive diagnosis of MTHFR deficiency in two NBS positive newborns (Pt 1, 3 Table 1) and in patient 5, born before the introduction of extended NBS (Pt 5, Table 1). Segregation analysis confirmed that the two mutated alleles were in *trans* (Figure 3) in all affected patients. Patients 1 and 5 showed, also, the c.665C>T polymorphism, in compound heterozygosity with the c.1286A>C polymorphism or in homozygosity, respectively (Table 1).

Patients 3 and 5 had known pathogenic variants (Table 1). Of note, patient 3 carried the synonymous c.1320G>A variant, which is predicted to alter the splicing process and cause a premature termination of translation [15].

Patient 1 was compound heterozygous for two missense variants, namely c.176G>C p.(Trp59Ser) and c.1769T>G p.(Leu590Arg). The first is a known disease-causing mutation that affects an amino acid residue matching to the enzyme’s catalytic domain (Figure 3D). In contrast, the c.1769T>G, p.(Leu590Arg), substitution, which maps in the regulatory region of the protein (Figure 3D), has not previously been reported in literature as associated to MTHFR deficiency. The c.1769T>G variant was absent from any of the following databases of public mutations and genetic variants databases: HGMD (http://www.hgmd.cf.ac.uk/ac/index.php, accessed on 14 December 2022) [37]; ClinVar (https://www.ncbi.nlm.nih.gov/clinvar/, accessed on 14 December 2022) [38]; National Center for Biotechnology Information (NCBI) (https://www.ncbi.nlm.nih.gov, accessed on 14 December 2022); 1000 Genome Project (http://www.internationalgenome.org/, accessed on 14 December 2022); Exome Sequencing Project (ExAC, http://evs.gs.washington.edu/EVS/, accessed on 14 December 2022) [39,40,41]; and Exome Genome Aggregation Database (http://gnomad.broadinstitute.org, accessed on 14 December 2022). We used the ACMG guidelines [34] to establish pathogenicity of the novel c.1769T>G variant identified in the MTHFR gene. In silico computational analysis predicted a pathogenic effect, with a high level of confidence, the substitution being classified as “disease causing”. Finally, segregation analysis showed that the variant c.1769T>G (paternally inherited) was in *trans* with the known pathogenic one (c.176G>C, maternally inherited). In agreement with these criteria, the novel c.1769T>G p.(Leu590Arg) variant can be classified as likely pathogenic. Interestingly, parents’ molecular analysis indicated that the c.1769T>G substitution was in *cis* with the c.665C>T polymorphism, whereas c.176G>C was in *cis* with c.1286A>C common variant.

We also predicted the secondary structures of MTHFR WT and MTHFR-Leu590Arg proteins, using the I-TASSER server. Both WT and mutant structures had the same structural similarity to the top ten proteins from the PDB that I-TASSER selected. The superimposed structures performed by PyMol showed a perfect match between WT and Leu590Arg proteins (Figure 4), with a very low RMSD (root-mean-square deviation of atomic positions) value of 1.143. Moreover, I-TASSER reported similar ligand binding target proteins for both wild type and mutant MTHFR proteins.

The other two NBS positive newborns (Patients 2 and 4, Table 1) carried the c.665C>T polymorphism in compound heterozygosity with the c.1286A>C polymorphism or the c.1160G>A mutation, respectively. In these subjects, we performed a quantitative analysis of the NGS data (coverage) relative to the exon regions of MTHFR gene and, in all cases, we obtained DI between 0.87 and 1.2, thereby excluding the presence of MTHFR gene macrodeletions.

## 4. Discussion

Deficiency of 5,10-methylenetetrahydrofolate reductase (MTHFR) is an autosomal recessive disorder affecting the remethylation of homocysteine into methionine. In fact, the MTHFR enzyme supplies the endogenous substrate of the methionine synthase enzyme, i.e.**,** the 5-MTHF, a critical methyl donor source in the remethylation of homocysteine into methionine. MTHFR deficiency can occur in neonatal or adolescence/adult-onset form, showing varying severity of disease, ranging from neonatal lethal to adult onset. However, the disorder is particularly responsive to early folate and betaine treatment; notably, treatment with mefolinate but not folic acid or folinic acid is effective to increase the methyltetrahydrofolate levels in the liquor, possibly preventing neurological damage (50). Indeed, promptly diagnosed and treated patients show a better clinical outcome with less mortality or severe organ complications [42,43,44]. Therefore, methylenetetrahydrofolate reductase deficiency is included in most newborn screenings worldwide [4]. From 2016, in Italy, the NBS comprises the screening for 47 hereditary metabolic diseases, including MTHF deficiency.

Generally, following the reduced Met level at the first test of screening and the increase of Hcy at the second tier tests, the definitive diagnosis takes advantages from molecular investigation, which identifies the genetic variations responsible for the disease, allows the couple to have genetic counselling and, as for other inherited diseases, access to prenatal diagnosis [45,46,47].

In this paper, we present the results of metabolic and genetic analysis of four newborns suspected to have MTHFR deficiency after NBS results, during 6 years (2017–2022) following the introduction of extended newborn screening in Italy. Within the study period, 249,642 newborns were screened. Of note, the introduction of the second-tier homocysteine in 2019 significantly reduced the false-positive rate for hypomethioninemia, which decreased from 0.028% (pre-second-tier homocysteine) to 0.00055% (post-second-tier homocysteine).

One patient, born before the introduction of extended NBS, was recruited based on neurological symptoms and metabolic investigation, showing increased plasma homocysteine levels.

In all cases a panel of biochemical investigations, including vitamin B12 levels, propionylcarnitine, and urinary methylmalonic acid was performed to consider differential diagnosis. The values of folate, medium corpuscular volume, acylcarnitines (in particular, the propionylcarnitine) were in the reference range in all patients at the first evaluation. Normal levels of methylmalonic acid and vitamin B12 allowed to exclude the diagnosis of methylmalonic acidemia due to intracellular synthesis defects of vitamin B12, and also a maternal deficiency of vitamin B12 in three out of four patients; patient 4 showed increased levels of MMA in combination with extremely low-levels of serum vitamin B12, both in the infant and in her mother, suggesting a maternal vitamin B12 deficiency.

In three patients, molecular analysis unveiled two pathogenic/likely pathogenic variants in the *MTHFR* gene, compatible with the definitive diagnosis of severe MTHFR deficiency. Two of them were identified by NBS, showing a cumulative incidence of this metabolic disorder in Campania Region of 2:249,642 newborns in six years. National data reported an incidence of 1:268923 [7]. For the third affected patient, an adolescent born in the pre-screening era with a history of psychomotor development delay and hyperhomocysteinemia, the molecular analysis was instrumental in reaching the definitive diagnosis.

Two of the three patients who tested positive to the molecular analysis were compound heterozygous for known pathogenic variants (3 and 5 in Table 1); on the contrary, patient 1 was a compound heterozygous with one allele carrying the novel likely pathogenetic variant c.1769T>G, p.(Leu590Arg). In particular, patient 1 carried on the other allele the known mutation c.176G>C, p.(Trp59Ser), which falls within the catalytic N-terminal domain of the MTHFR enzyme. Pathogenic changes in this domain usually affect the NADPH binding and cause MTHFR deficiency. In agreement, this variant is associated with a severe decrease in enzymatic activity, resulting in early onset of symptoms and death within the first year of life [17]. The new variant c.1769T>G, p.(Leu590Arg), identified in patient 1 was classified as likely pathogenetic according to the ACMG criteria [34]. It falls within the regulatory domain of the enzyme, which likely contributes to NADPH binding and includes the binding site for SAM, the allosteric inhibitor of MTHFR. As mutations in the regulatory domain can impair the binding with SAM and alter the mechanism of inhibition of the MTHFR enzyme [15,48], we used a bioinformatics approach to predict a three-dimensional structure model of MTHFR WT and MTHFR Leu590Arg proteins, and did not identify significant differences in the overall structure by comparing the WT and MTHFR Leu590Arg models. However, the modelling highlighted that Leu590, a well-conserved residue across distant species such as mammals and even zebrafish D. Rerio, is buried in MTHFR structure. Moreover, it is encoded by the exon 11, which is a hot-spot of mutations that severely affect the MTHFR regulatory region [17]. The replacement of the apolar leucine with a positively charged arginine could destabilize the high ordered three-dimensional structure of the molecule, leading to its unfolding and inactivation.

In addition to pathogenetic/likely pathogenetic variants, patients 1 and 5 also showed a compound heterozygosity for the two well studied polymorphisms [16] c.665C>T p.(Ala222Val) [MAF (Minor allele frequency) 30%)] and c.1286A>C p.(Glu429Ala) (MAF: 29%), or homozygosity for the c.665C>T polymorphism, respectively. As in other diseases, polymorphisms may have functional effects [49]. In particular, c.665C>T and c.1286A>C polymorphisms cause a mild reduced MTHFR activity and are known genetic risk factors in thrombophilia, either in the homozygous or compound heterozygous state [20]. In particular the c.665C>T polymorphism encodes a thermolabile MTHFR enzyme, due to reduced FAD binding, resulting in dissociation of the holoenzyme into active monomers [50,51]. The polymorphism is associated with 30% and 65% of residual enzyme activity in homozygotes and heterozygotes, respectively, and with mildly elevated plasma homocysteine levels [20]. The presence on the same allele of the c.665C>T polymorphism and a deleterious mutation further decreases the residual MTHFR activity respect to those caused by the mutation alone [15]. Our patients show the c.665C>T polymorphism in *cis* with the novel variant c.1769T>G (patient 1), or in *cis* with the known mutations c.973C>T and c.1970G>C (patient 5), suggesting a possible synergy between these mutations and the polymorphism.

The two patients diagnosed in the neonatal period, according to the most recent and accepted guidelines, started a therapy with betaine, 5-methyltetrahydrofolate and variable parenteral hydroxocobalamin (which was discontinued at three months in patient 1). In fact, early treatment with a combination of high dose mefolinate and betaine prevents mortality and allows normal psychomotor development in patients with severe MTHFR deficiency [42,43,44,52]. Their clinical monitoring, over 24–48 months (patient 3 and 1, respectively), showed adequate weight gain and no signs and/or symptoms associated with the disease. Patient 5 was diagnosed after acute neurological episodes and started prompt therapy with 5-methyltetrahydrofolate and vitamin B complex, with only partial homocysteine response. The therapy was then implemented with betaine supplementation after evaluation in our metabolic unit with further drop in homocysteine values. The clinical improvement has been slower. The patient has regained most of her cognitive ability and personal autonomy but lower limb spasticity and walking difficulties remained. However, no other acute neurological episode has been described. The clinical outcome of patient 5 clearly demonstrates the importance of initiating therapy in the pre-symptomatic period to prevent disease complications and progression of the damage.

Genetic counselling was offered to all families. Family segregation showed that all relatives of probands carried one mutation in the *MTHFR* gene; in a subsequent pregnancy, one at-risk couple also required prenatal diagnosis, which diagnosed the presence of a fetus heterozygous for the disease. Furthermore, family genotyping also identified additional subjects who were carriers of a genetic risk factor for thrombophilia.

The other two neonates with positive NBS results were not affected by severe MTHFR deficiency: they were compound heterozygous for the two polymorphisms c.665C>T and c.1286A>C, and the c.665C>T polymorphism and a known mutation in the *MTHFR* gene, respectively. These genotypes determine a mild reduction in enzyme activity that can be unveiled by newborn screening. In addition, patient 4 suffered of maternal vitamin B12 deficiency, which likely contributes to extremely high hyperhomocysteinemia in combination with high excretion of methylmalonic acid. However, considering the initial Hcy value and the small possibility that sequencing analysis missed the variant on the second allele, paediatricians continued the clinical follow-up of patient 4 for three months before losing the patient to follow up. Last evaluation of homocysteine level after hydroxcobalamin therapy showed normal value. We concluded for a diagnosis of vitamin B12 deficiency, from maternal origin, but the role of the heterozygous *MTHFR* gene variant in the neonatal homocysteine elevation cannot be excluded.

## 5. Conclusions

In summary, this paper presents the first descriptions in Europe of MHTFR variants detected after implementation of the expanded NBS. Second tier tests detecting hyperhomocysteinemia are widely applied with good results. However, accurate and timely molecular diagnosis is a key aspect to achieve the definitive diagnosis of MTHFR deficiency, after a positive result at NBS and it is crucial for clinical management and therapeutic choices. In fact, the definitive diagnosis allows making targeted therapeutic choices, with high-dose mefolinate and betaine, which may improve outcome in pre-symptomatic patients [53]. In addition, genetic testing is instrumental to reach definitive MTHFR deficiency diagnosis also in symptomatic adolescence/adult patients. These patients may manifest isolated neurological/psychiatric symptoms, which may delay the achievement of the diagnosis; however, metabolic treatment may stabilize or improve symptoms [11]. The key role of genetic testing in reaching the definitive diagnosis opens the door to a possible “genotype-first” approach in newborns/children presenting with hyperhomocysteinemia. In the era of precision medicine, molecular analysis of a targeted genes panel associated with hyperhomocysteinemia, including *MTHFR* gene, should be the first-tier genetic test for newborns/children suspected to have an inherited remethylation disorders. Furthermore, genetic diagnosis allows identification of carriers within families: this issue is very important to identify subjects having an increased risk of venous thromboembolism, also. Finally, the identification of a novel pathogenic gene variant enlarges the mutation spectrum of the *MTHFR* gene, although most mutations in the *MTHFR* gene are private.

Our data confirm that the incidence and prevalence of this rare disease are very low also in the population of the Campania region, in accordance with national data [7].

Finally, the results here presented reflect the benefits of close collaboration among screening, confirmation laboratories and metabolic pediatricians for the efficacy of newborn screening programs.

## Figures and Tables

**Figure 1 genes-14-00980-f001:**
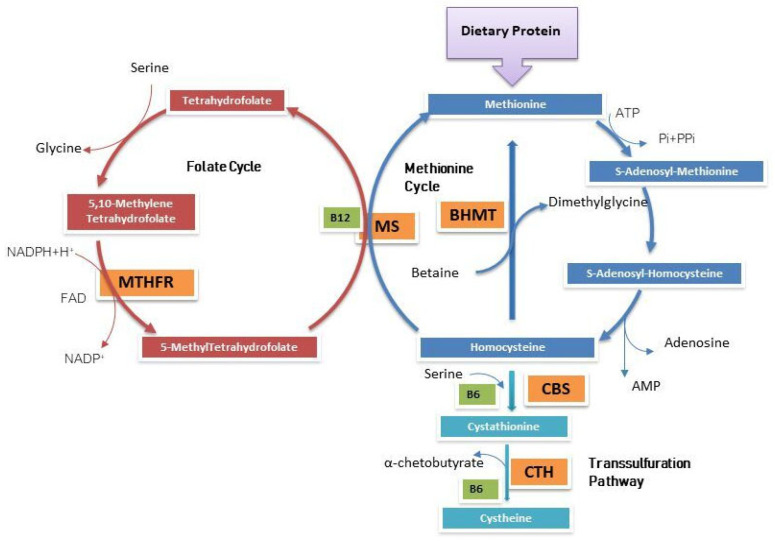
Methionine’s methylation pathway. *BHMT*: Betaine-Homocysteine Methyltransferase; *CBS*: Cystathionine **β**-Synthase; *CTH*: Cystathionine **γ**-Lyase; *MS*: Methionine synthase; *MTHFR*: 5,10-Methylenetetrahydrofolate reductase.

**Figure 2 genes-14-00980-f002:**
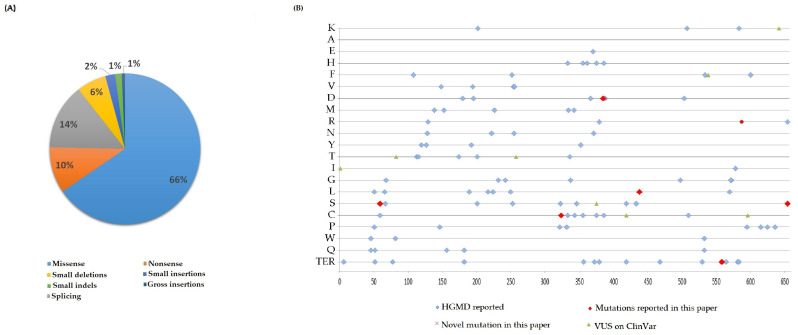
(**A**) MTHFR gene mutations reported in HGMD**^®^** Professional 2022.4, according to type of mutation. (**B**) The graph represents the localization of the mutations along the MTHFR protein. We include variants reported on ClinVar as VUS (variant of uncertain significance), but currently classified as likely pathogenetic according to ACMG criteria.

**Figure 3 genes-14-00980-f003:**
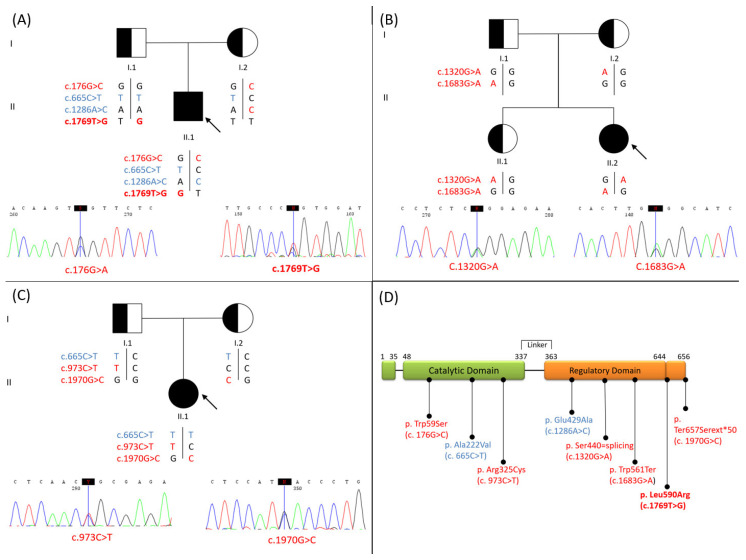
MTHFR patients’ pedigree. (**A**) Patient 1. (**B**) Patient 3. (**C**) Patient 5. (**D**) The figure is a schematic representation of MTHFR protein domains (catalytic and regulatory), showing the localization of mutations identified in patients. Mutations are in red (the novel one in bold). The two common polymorphisms (p.Ala222Val and p.Glu429Ala) are shown in blue. 
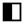
 Carrier male; 
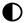
 Carrier female; 
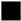
 Affected male; 
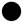
 Affected female; 
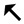
 Proband.

**Figure 4 genes-14-00980-f004:**
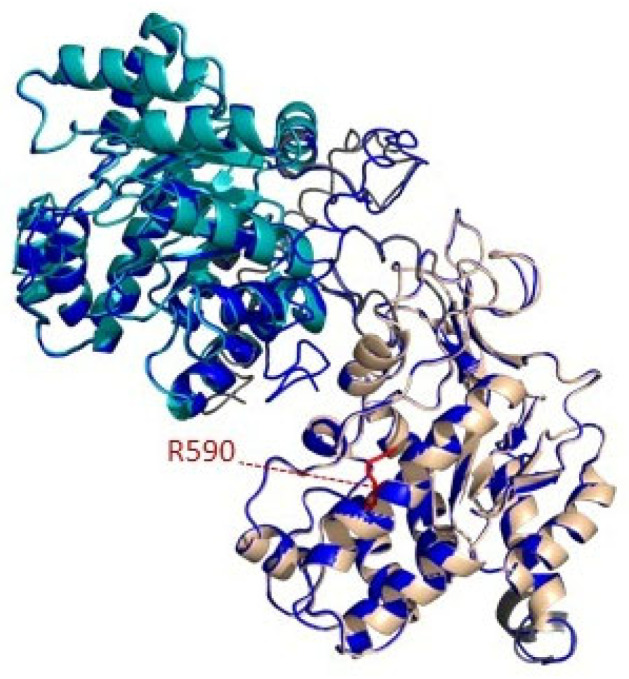
Homology modelling prediction of MTHFR-L590R. Graphic representation of secondary structure of MTHFR mutated protein, modelled with I-tasser using X-ray structure of human MTHFR as template (6FCX PDB structure). The structure is composed by catalytic domain (represented in cyan), regulatory domain (represented in wheat) and the linker between them (represented in gray). The position of the arginine replacing the wild-type amino acid in patient is shown in red. The models of MTHFR have been prepared with PyMOL.

**Table 1 genes-14-00980-t001:** Biochemical and molecular results detected in five patients suspected for MTHFR deficiency in years range 2017–2022.

	Biochemical Markers						Molecular Analysis				
Patient (Date of Birth)	*DBS (I Spot)*		*DBS (II Spot)*		*Serum Tests*		HGVS ^§^ cDNA	HGVS ^§^ Protein	Variant Classification		
	Met *	Hcy ^#^	Met *	Hcy ^#^	Met **	Hcy ^#^			ClinVar	HGMD ^§§^	ACMG ^§§§^
**Patient 1** (20 February 2018)	4	53.7	6	42.3	10	106.7	c.176G>C c.665C>T c.1286A>C **c.1769T>G**	p.Trp59Ser p.Ala222Val p.Glu429Ala **p.Leu590Arg**	P LB (DR) CI **NR**	CM155512 CM950819 CM981315 **NR**	LP B B **LP**
**Patient 2** (7 February 2020)	5.5	4.4	9	5.11	31	14.7	c.665C>T c.1286A>C	p.Ala222Val p.Glu429Ala	LB (DR) CI	CM950819 CM981315	B B
**Patient 3** (20 October 2020)	5.07	11	8.6	16.3	29	53.1	c.1320G>A c.1683G>A	p.Ser440 = (splicing) p.Trp561Ter	P/LP P	CM155527 CM155520	LP P
**Patient 4** (24 July 2022)	5.6	9.2	8	7	23	54.5	c.665C>T c.1160G>A	p.Ala222Val p.Gly387Asp	LB (DR) NR	CM950819 CM000527	B VUS
**Patient born before the introduction of NBS**											
**Patient 5** (1 August 2009)	NP	NP	NP	NP	NP	423 pre therapy 27.9 during therapy	c.665C>T c.973C>T c.1970G>C	p.Ala222Val p.Arg325Cys p.Ter657Serext*50	LB (DR) VUS LB	CM950819 CM950822 CM035841	B LP LP

*: lower cutoff 6 μmol/L; ^#^: upper cut-off: 4 μmol/L; **: rv 10–60 μmol/L (<1 month), 9–42 μmol/L (>1 month); ^§^: human sequence variant nomenclature; ^§§^: human genetic mutation database; ^§§§^: American College of Medical Genetics and Genomics classification; DBS: dry blood spot; Met: methionine; Hcy: homocysteine; NP: not performed; NR: not reported; LP: likely pathogenic; P: pathogenic; LB: likely benign; VUS: variant of uncertain significance; CI: conflicting interpretation; B: benign; DR: drug response; NBS: newborn screening. Pathologic data are in red. Novel mutation in bold. All variants are at heterozygous state except the homozygous c.665C>T detected in patient 5.

## Data Availability

All research data related to NBS, molecular analysis and clinical management of patients reported in the paper can be requested from the corresponding authors.

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
