# Peer review of "Contribution of Genetic Test to Early Diagnosis of Methylenetetrahydrofolate Reductase (MTHFR) Deficiency: The Experience of a Reference Center in Southern Italy"

_genes, 2023, doi:10.3390/genes14050980_

Round 1

Reviewer 1 Report

This manuscript describes newborn screening and confirmatory diagnoses of MTHFR deficiency. The condition causes severe neurological damages that can be prevented by early detection and administration of drugs such as betaine. However, newborn screening for this potentially preventable disease has not been established, which is in part due to the difficulty in detecting affected patients by hypomethioninemia with enough sensitivity and specificity. Therefore, the data of pilot study on newborn screening for diseases with hypomethioninemia will be  useful for those who are interested in newborn screening for these disorders. My questions and comments are as follows.

1)

The method of newborn screening should be described more mimutely, including the cutoff value, not the reference range, of methionine and total homocysteine in the first DBS, and explain how the cutoff values were set (based on a preliminary study on positive rate, I suppose).

2)

How many newborns were screened within the study period (if it is 206548, the value should be clearly shown in Results, not in Discussion only) and how many of them were positive for hypomethioninemia? False-positive rate without 2nd-tier homocysteine is important to evaluate the screening from practical point of view.

3)

Table 1 is confusing at a glance because patient 4 shows marked elevation of total homocysteine in spite of heterozygous disease-causing mutation. The table will be more informative by adding data of C3 in DBS, vitamin B12 in serum, and MMA in urine.

4)

Though characterizing the mutations is of scientific interest, clinical information such as the age at start of therapy, dosage of each drug, time-course of methionine and total homocysteine will be more useful to practitioners reading this article.

Author Response

This manuscript describes newborn screening and confirmatory diagnoses of MTHFR deficiency. The condition causes severe neurological damages that can be prevented by early detection and administration of drugs such as betaine. However, newborn screening for this potentially preventable disease has not been established, which is in part due to the difficulty in detecting affected patients by hypomethioninemia with enough sensitivity and specificity. Therefore, the data of pilot study on newborn screening for diseases with hypomethioninemia will be  useful for those who are interested in newborn screening for these disorders. My questions and comments are as follows.

1) The method of newborn screening should be described more mimutely, including the cutoff value, not the reference range, of methionine and total homocysteine in the first DBS, and explain how the cutoff values were set (based on a preliminary study on positive rate, I suppose).

 According to the referee’ s suggestion, we described the method of newborn screening and how the cutoff values have been set (lines 121-129). We apologize for the error in table 1, which reported the “reference value” of methionine and total homocysteine in the DBS. We have now replaced the reference value with the cutoff values (Table 1, line 265)

2)How many newborns were screened within the study period (if it is 206548, the value should be clearly shown in Results, not in Discussion only) and how many of them were positive for hypomethioninemia? False-positive rate without 2nd-tier homocysteine is important to evaluate the screening from practical point of view.

We reported the number of newborns screened within the study period also in the Result section (line 188-189). The correct number is 249,642. We apologize for the mistyping. Furthermore, the false-positive rate without 2nd-tier homocysteine was highlighted (lines 359-362).

3)Table 1 is confusing at a glance because patient 4 shows marked elevation of total homocysteine in spite of heterozygous disease-causing mutation. The table will be more informative by adding data of C3 in DBS, vitamin B12 in serum, and MMA in urine.

and

4)Though characterizing the mutations is of scientific interest, clinical information such as the age at start of therapy, dosage of each drug, time-course of methionine and total homocysteine will be more useful to practitioners reading this article.

 We accepted the referee's suggestion and added a supplementary table (ST1),  showing, for patients 1-4, the requested additional data, both related to the biomarkers and their time-course, both related to the therapeutic regimen. Treatment regiment for patient 5 was added in the text (line 247).

Reviewer 2 Report

This manuscript has several strengths. Overall, it is presented well; the authors have presented the data clearly. The organization, flow, and readability of the paper are good. The authors characterized the patients and the disease in question comprehensively. The language, other than the need for some minor English editing, is engaging and clear. The manuscript will also need some revision, however, to address the following issues. The central argument was difficult to discern at times. What specific question were the authors answering? The paper may benefit from making this clearer. Additionally, providing some idea of future directions and the potential impact (e.g. what does this change, and where do we go from here?) would likely help the readers understand the importance of this research. Finally, the context of this work should be explored in greater detail. In the Conclusion, the authors include the phrase “this paper presents the first descriptions in Europe of MHTFR variants detected after implementation of the expanded NBS,” which helps define the context and impact of the manuscript. However, to make this statement, the authors need to provide a clear description of what has already been done, and what is known in this area. The Introduction would be an appropriate place to include this information.

Overall, this is a strongly written paper in need of some revision.

Author Response

This manuscript has several strengths. Overall, it is presented well; the authors have presented the data clearly. The organization, flow, and readability of the paper are good. The authors characterized the patients and the disease in question comprehensively. The language, other than the need for some minor English editing, is engaging and clear. The manuscript will also need some revision, however, to address the following issues. The central argument was difficult to discern at times. What specific question were the authors answering? The paper may benefit from making this clearer. Additionally, providing some idea of future directions and the potential impact (e.g. what does this change, and where do we go from here?) would likely help the readers understand the importance of this research. Finally, the context of this work should be explored in greater detail. In the Conclusion, the authors include the phrase “this paper presents the first descriptions in Europe of MHTFR variants detected after implementation of the expanded NBS,” which helps define the context and impact of the manuscript. However, to make this statement, the authors need to provide a clear description of what has already been done, and what is known in this area. The Introduction would be an appropriate place to include this information.

Overall, this is a strongly written paper in need of some revision.

We thank the referee for the constructive suggestions. We have modified the Introduction according to the indications of the referee (lines 80-94) and we have discussed in the Conclusions the potential impact of our results (lines 491-496).

These changes have certainly made the paper more interesting to the readers.